



# Salinity dynamics of the Baltic Sea

Andreas Lehmann[1], Kai Myrberg[2,3], Piia Post[4], Irina Chubarenko[5], Inga Dailidiene[6], Hans-Harald Hinrichsen[1], Karin Hüssy[7], Taavi Liblik[8], Urmas Lips[8], H. E. Markus Meier[9], Tatiana Bukanova[5]

[1] GEOMAR Helmholtz Centre for Ocean Research Kiel, Germany
[2] Finnish Environment Institute/Marine Research Centre Helsinki, Finland
[3] Marine Research Institute, Klaipeda University, Klaipeda, Lithuania
[4] Institute of Physics, University of Tartu, Estonia
[5] Laboratory of Marine Physics, P.P.Shirshov Institute of Oceanology RAS, Kaliningrad, Russia
[6] Faculty of Marine Technology and Natural Sciences, Klaipeda University, Klaipeda, Lithuania
[7] National Institute of Aquatic Resources, Technical University of Denmark, Charlottenlund, Denmark

[8] Marine Systems Institute at Tallinn University of Technology, Tallinn, Estonia
[9] Leibniz Institute for Baltic Sea Research, Warnemünde, Rostock, Germany

*Correspondence to*: Andreas Lehmann (alehmann@geomar.de)

**Abstract.** In the Baltic Sea, salinity and its large variability, both horizontal and vertical, are key physical factors in determining the overall stratification conditions. In addition to that, salinity and its changes also have large effects on various ecosystem processes. Several factors determine the observed two-layer vertical structure of salinity. Due to the excess of river runoff to the sea, there is a continuous outflow of water masses in the surface layer with a compensating inflow to the Baltic in the lower layer. Also, the net precipitation plays a role in the water balance and consequently in the salinity dynamics. The salinity conditions in the sea are also coupled with the changes in the meteorological conditions. The ecosystem is adapted to the current salinity level: a change in the salinity balance would lead to ecological stress of flora and fauna, and related negative effects on possibilities to carry on sustainable development of the ecosystem. The Baltic Sea salinity regime has been studied for more than 100 years. In spite of that, there are still gaps in our knowledge of the changes of salinity in space and time. An important part of our understanding of salinity are its long-term changes. However, the available scenarios for the future development of salinity are still inaccurate. We still need more studies on various factors related to salinity dynamics. Among others more knowledge is needed, e.g. from meteorological patterns in various space and time scales and mesoscale variability in precipitation. Also, updated information on river runoff and inflows of saline water is needed to close the water budget. We still do not understand accurately enough the water mass exchange between North Sea and Baltic Sea and within its sub-basins. Scientific investigations of the complicated vertical mixing processes are additionally required. This paper is a continuation and update of the BACC II book which was published in 2015, including information from articles issued until 2012. After that, there have been many new publications on the salinity dynamics, not least because of the Major Baltic Inflow which took place in December 2014. Several key topics have been investigated, including the coupling of long-term variations of climate with the observed salinity changes. Here the focus is on observing and indicating the role of climate change for salinity dynamics. New results of MBI-dynamics and related water mass



interchange between the Baltic Sea and the North Sea have been published. Those studies also included results from the MBI-related meteorological conditions, variability in salinity and exchange of water masses between various scales. All these processes are in turn coupled with changes in the Baltic Sea circulation dynamics.

# 1 Introduction

The Baltic Sea salinity is not only a physical variable, but it also describes in an integrated way the simultaneous effects of the energy and water cycles in the sea; some of these features are just typical for the Baltic Sea, as the low mean level of salinity and its pronounced variability. Several factors determine the observed structure of salinity. Due to the excess of river runoff to the sea, there is a continuous outflow of water masses in the surface layer. A compensating inflow to the Baltic Sea takes place from the Kattegat through the Danish Straits in the lower layer, strongly governed by the local atmospheric conditions. Also, the net precipitation over the sea plays a role in the water balance and consequently in the salinity dynamics. An essential role in salinity dynamics is played by the barotropic water exchange which comprise irregularly Major Baltic Inflows (MBIs, Matthäus and Franck, 1992) and Large Volume Changes (LVCs, Lehmann et al., 2017). These inflows have a significant impact on the modification of the observed patterns of stratification and oxygen conditions.

The Baltic Sea salinity regime has been studied for more than 100 years. Despite this long research history, there are still gaps in our knowledge of salinity changes, both in space and time. Due to that, the available scenarios for the future are still inaccurate. The projections indicate that precipitation will increase during the forthcoming decades. Hence, the ensemble mean of available scenarios shows a decrease of salinity by about 0.6 g kg$^{-1}$ until 2100. The global rise of sea level has not been taken into account in these assumptions for the future (Saraiva et al., 2019b). Due to the uncertainties in water balance estimates, there are inaccuracies in the climatic model scenarios: whether the Baltic Sea salinity will decrease, or increase is still an open question. The Baltic Sea ecosystem is adapted to the current salinity level: a change in the salinity balance would lead to ecological stress of flora and fauna and related negative effects on possibilities to carry on sustainable development of the ecosystem (e.g. Vuorinen et al., 2015).

BACC II book (BACC II Author Team 2015; Elken et al. 2015) includes the review of the salinity dynamics based on publications till 2012. After that, especially the Baltic Earth community has encouraged scientist to publish new results on that issue. In December 2014, a Major Baltic Inflow took place and afterwards, several papers were devoted to study various aspects of such inflow events (see Mohrholz, 2015, Gräwe et al. 2015, Rak, 2015). Those studies revealed new results on multiple factors not only concerning MBIs. Such are: the link between long-term (decadal-scale) variability in climatic conditions with the salinity development in the Baltic Sea, MBIs and related barotropic exchange of mass and meteorological forcing conditions, variations in salinity and fluxes on various scales (observation and attribution to changes in climate), salt budget changes and the related variations in the Baltic Sea circulation and induced changes in oxygen conditions.





This paper is organized as follows. First, we summarize the knowledge which has been collected and summarized in BACC I (BACC Author Team, 2008) and BACC II (BACC II Author Team, 2015) books and e.g. in Leppäranta and Myrberg (2009) and Omstedt et al. (2014). Additionally, we assess recent publications and knowledge following the BACC-process after 2012. This part starts with describing the atmospheric forcing which is driving the salinity dynamics, followed by a detailed update of the knowledge of salinity dynamics. Further on, we study new features of salinity dynamics on regional scale concerning the sub-basins surrounding the Baltic Proper (Fig. 1). The various sub-basins respond differently to the changing atmospheric conditions. So, we highlight observed similarities and differences. Further, we summarize climate change impact on salinity dynamics. Following that, oxygen conditions are analyzed in the central deep areas, being directly related to the dynamics of salinity. Thus, an improved understanding of the salinity dynamics will also deepen our knowledge of the processes concerning oxygen. Additionally, the salinity dynamics is also related to the environmental conditions of the marine ecosystem, like fisheries, etc., in the Baltic Sea, which is discussed, too. The paper ends with discussing existing knowledge gaps and by giving key messages of our present understanding of salinity dynamics and suggesting necessary further work.

## 2. Salinity dynamics of different space and time scales – knowledge from BACC I and BACC II

Salinity dynamics have been discussed in both BACC books (BACC I Author team, 2008; BACC II Author Team, 2015) and, e.g. in Leppäranta and Myrberg (2009) and Omstedt et al. (2014). We will summarize here the earlier findings of salinity dynamics to set up the basis of our current understanding:

- There was a decreasing trend of the mean salinity of the Baltic Sea both in the early 1900s and later during the century (1980s and 1990's); the latter is coupled with a complete lack of MBIs during 1983-1993. During those periods, freshwater inflows were more extensive than on average and zonal winds were stronger than normal, showing a very long-term natural variability in the highly dynamic system. During the stagnation periods with lower than normal salinities, the lack of inflows led to the situation where the ventilation in the Gotland basin was weak below the halocline. As a consequence of that, hypoxic bottom areas formed. However, towards the end of the period without inflows, the hypoxic area was shrinking because of the deepening and weakening of the halocline. For example, in the Gulf of Finland, the halocline completely disappeared, and the bottom oxygen conditions improved. However, despite the abrupt changes in salinity, there was no clear trend for the vertical mean salinity, if to consider the entire 20th century.

- During the last 2-3 decades, the surface-layer salinity was slightly lower than on average, supposed to be driven by higher accumulated river runoff.

- MBIs, usually of barotropic origin, occur in favorable meteorological conditions, such as mainly existing in winter and springtime. For an MBI to happen, there should be firstly winds from the east, and after such a period,





winds should blow from the west for several weeks. The back-to-back occurrence of these two wind events is not

very common in the Baltic Sea, which keeps the natural frequency of MBIs relatively low.

- Later, after 1996, a different type of inflow has been observed. Such events are baroclinicall-driven and take place during the summer period. In the same way as for barotropic events such inflows transport water with higher salinity and temperature to the Baltic deeper layers. But, as the inflowing water volume of higher salinity is small compared to MBIs, the water stratifies in the halocline, not able to substitute the bottom water. Summer

105       inflows inject higher saline water with higher temperatures and low oxygen content into the halocline. Most probably, such events have occurred before but could not be observed due to shortcomings in the observational strategy.

## 3. Atmospheric forcing driving the salinity dynamics of the Baltic Sea

The large-scale atmospheric circulation controls the local weather conditions over the Baltic Sea area, which in turn drives

the circulation in the Baltic Sea and the distribution of temperature, salinity and oxygen, which are relevant for biological production. Different weather regimes impact on the trophic structure and the marine food webs (Lehmann et al. 2002; Hinrichsen et al. 2007).

The speed and position of the Atlantic storm track or the polar jet stream is the most prominent feature that influences the variability of the atmospheric conditions in the Baltic Sea region. The influence of this storm track could be described in

various space and time scales. The mean salinity of the Baltic Sea is controlled by long-term variations such as river runoff, dry and wet periods. At the same time, salinity variations on smaller scales are driven by shorter-term events like barotropic exchange flows. Starting from the largest, the continental or hemispheric scale, teleconnection patterns are commonly used to describe the atmospheric circulation variability. The best-known of them is the North Atlantic Oscillation (NAO), which is the first mode of principal component analysis of the sea level pressure (SLP) field over the North Atlantic/European

sector (Hurrel, 1995). The East Atlantic (EA) pattern (Wallace and Gutzler, 1981) and the Scandinavian pattern (SCA), also termed Eurasian or blocking pattern, are described by the second and third mode, respectively (Hurrel and Deser, 2009). All these modes are better expressed in winter than in other seasons.

The NAO index or similar local indices (e.g. BSI – Baltic Sea Index, Lehmann et al., 2002), which describe the strength of the zonal atmospheric circulation, are often related to the intermittent water exchange between the North Sea and Baltic Sea

through the Danish Straits. The popularity of the NAO resulted from its relatively close connection with the decadal variability of the seasonal circulation in the years 1960-1990 when the NAO index increased, and the correlation between the climatological variables in Northern Europe (including the Baltic Sea surface elevation) was very high (Pinto and Raible, 2012; Feser et al., 2015; Lehmann et al., 2017). But the link is non-stationary, and therefore this simple approximation does not work for all periods (Cassou et al., 2004; Matulla et al., 2008; Lehmann et al., 2017). The analysis of winter SLP data

highlighted considerable changes in intensification and location of storm tracks, parallel with the eastward shift of the NAO





centres of action (Cassou et al., 2004; Lehmann et al., 2011). At the same time, a seasonal shift of extreme wind events from autumn to winter and early spring was found in the Baltic area.

The strength of windstorms is undoubtedly crucial for the salinity dynamics of the Baltic Sea. Zubiate et al. (2016) characterized the variability of wind speed and distribution as a function of the NAO and the current states of the secondary

(EA) and tertiary (SCA) patterns of the SLP variability over Europe. A strong correlation at monthly time scales has been found between the NAO positive phase and wind speed in northern Europe. But this effect combines with different other patterns that vary with sub-region. Over the Danish Straits, strong winds are associated with the combination of the NAO$^+$, EA$^-$ and SCA$^-$ phase, while over Scandinavia, the NAO$^+$ combined with EA$^+$ initiates more storms. The temporal clustering of windstorms, also an essential player in wind climatology, has different large-scale drivers dominating over the Danish

Straits. There is a triple point of NAO, SCA, and the polar index (POL), with POL dominating in the northern flank and the SCA over the Baltic Sea's southern side (Waltz et al., 2018). All this refers that the Baltic Sea region is not homogeneous from the viewpoint of large-scale atmospheric variability. Thus, it is essential to investigate forcing patterns also at smaller scales.

The analysis of the synoptic-scale atmospheric circulation is based on classifying meteorological fields with various methods

or tracking cyclones and anticyclones, mapping, and counting them (Barry and Carleton, 2013). Both kinds of approaches are applied to characterize the atmospheric conditions before, during, and after the events of large barotropic inflows or large volume changes (LVC, Lehmann et al., 2017). From earlier studies (e.g., Schinke and Matthäus, 1998), it could be deduced that the synoptic-scale atmospheric forcing, which is vital during inflow events, could be described and interpreted by the usage of automated weather types. Two different synoptic classifications have been applied (Lehmann and Post, 2015; Post

and Lehmann, 2016). During different phases of the inflow event, the number of certain classes (directions of synoptic-scale airflow) increases or decreases compared to the average frequency of classes. About 60 days before the maximum inflow, which corresponds to the maximum sea surface elevation at Landsort, the frequency of eastern and southeastern classes increases for about 30 days. This confirms the results of Matthäus and Schinke (1994) about the pre-inflow period with prevailing easterly winds and less precipitation to enhance the outflow of Baltic Sea water lowering the mean sea level. At

the same time, the wedge-shaped salinity front in the Danish Straits becomes more tilted by the movement of higher saline bottom water in the direction to the sill. An immediate period of very strong westerly winds starting about 30 days before the maximum inflow force effective LVCs/MBIs. Atmospheric forcing is more strongly associated with LVCs than MBIs, while it directly controls the sea level and indirectly the amount of salt of the inflowing water mass.

Barotropic inflow events like LVCs/MBIs are driven by a sequence or accumulation of atmospheric forcing (Lehmann et al.,

2017). During barotropic inflow events, which last about 40 days, 5–6 temporal clustered deep cyclones move along characteristic pathways or storm tracks.

One possible reason for less frequent MBIs in the 1980s might be the increased atmospheric zonal circulation associated with increased precipitation and runoff at the expense of pre-inflow easterly wind periods (Schinke and Matthäus, 1998; Lehmann et al., 2002; Meier and Kauker, 2003; Lehmann et al., 2011). Soomere et al. (2015) proposed an alternative





explanation. The meridional airflow direction over the southern Baltic Sea changed around 1987 to northwestern wind
       events at the expense of wind directions necessary for MBIs to occur.

## 4. Update of the knowledge of salinity dynamics since 2012

### 4.1 Large Volume Changes and Major Baltic Inflows

Despite a long history of Baltic Sea studies, still today, an important objective for investigations is the dynamics of water
       exchange between the North Sea and the Baltic Sea. A critical study objective is the transition area between the two seas: the
       Danish Straits.
       Furthermore, pronounced changes of the salinity in the Gotland Deep, situated in the central Baltic, are strongly dependent
       on major salt water inflows, MBIs These events represent a specific type of barotropically-driven inflows. Their formation
depends on specific favorable meteorological circulation patterns (e.g. Matthäus et al. 2008, Leppäranta and Myrberg, 2009;
       Lehmann and Post, 2015) and spatio-temporal grouping of deep cyclones (Lehmann et al., 2017).
       There has recently been carried out a number of investigations about the role of meteorological forcing and its impact on the
       hydrographic conditions in the Danish Straits, and the total freshwater supply to the Baltic Sea before the occurrence of
       highly saline barotropically-driven inflows. Höflich and Lehmann (2018) proposed a mechanistic explanation including the
salinity in the Danish Straits as well as the time-variability of meteorological forcing in connection to an inflow taking place.
       Freshwater supply played only a modulating role, i.e. it does not lead to a change in frequency nor intensity of the events.
       However, it had an adjusting role to observed transports of saline water to the Baltic Sea.
       The third-largest ever observed MBI took place in December 2014, rising interest to study the saline inflows further (e.g.
       Mohrholz et al., 2015, Gräwe et al. 2015, Rak, 2016, Neumann et al., 2017, Liblik et al., 2017). Mohrholz (2018) re-analysed
the time series of the Major Inflows. He used long-term data of sea levels, river runoff and observed salinity in the Belt and
       the Sound. As a result, an ongoing time series of barotropically-driven inflows was formed from about 1890 until today. The
       new time series by Mohrholz (2018) were compared with those composed by Fisher and Matthäus (1996). There were found
       apparent differences between the two series from the 1980s onwards. There were many reasons behind the observed
       deviations. Not much accurate data during the 1976-1991 period was available. After that time, the methods to observe the
inflows were changed, which caused bias to the statistics. Moreover, the locations where the measurements took place were
       changed (Mohrholz, 2018).
       There is a clear difference in the revised time series of the MBIs compared with the earlier ones. Namely, according to
       Mohrholz (2018), there is no clear trend in the frequency and intensity of the MBIs in the decadal time scale. In the
       traditional assumption, climate change would have impacted MBIs as decreasing trend in their frequency. On the other hand,
there was found to be variability in the frequency of the inflows with a timescale of about 30 years (Mohrholz, 2018 and
       Radtke et al., 2020).



This decadal variability was also found in surface and bottom salinities, river runoff and salt transport across the Darss Sill

(Radtke et al. 2020). It also turned out that MBIs are not the only events which transport salt into the Baltic Sea. There are

also smaller inflows of barotropic origin. These occur during all seasons having a low variability between the years. Such

inflows bring about 30 % of the entire salt transport to the Baltic Sea. This variability of the saline water inflows in time does

not explain the worsening of oxygen conditions in the Gotland basin (anoxic bottoms) and the observed prolonged periods of

stagnation (Mohrholz, 2018). Large barotropic inflows and the associated dense bottom currents form one branch of the

Baltic Sea overturning circulation and deep-water ventilation. Holtermann et al. (2017) investigated the dynamics of the deep

waters and vertical mixing in the central part of the Baltic Sea while Major Baltic Inflows took place; thus, providing new

information on dense bottom gravity currents on their way to the deep central Baltic Sea and associated turbulent mixing.

Liblik et al. (2018) studied the impact of MBIs downstream from the eastern Gotland Basin to the Gulf of Finland. A further

study by Stramska and Aniskiewicz (2019) showed that remote sensing altimetry could be a complementary source of

information about barotropic inflow events.


## 4.2 The cold intermediate layer

A Baltic-specific physical feature is the so-called Cold Intermediate Layer (hereafter denoted as CIL), which is formed

annually. CIL appears as a temperature minimum between the thermocline and the permanent halocline from spring to

autumn. Vertical convection, due to cooling of the atmosphere, and wind mixing erodes the seasonal thermocline during

autumn and winter (Leppäranta and Myrberg, 2009, Stepanova, 2017). During this process, the water masses of the fresher

upper layer and the sub-thermocline saltier layer are mixed. As a result, the seasonal salinity maximum in the surface layer

occurs in winter (Reissmann et al., 2009). This mixing process extends down to the upper boundary of the permanent

halocline (Fig. 2 and 3). In areas where the permanent halocline does not exist, such as the Gulf of Riga and the Gulf of

Bothnia (Fig. 1), mixing leads almost every winter to a complete turnover of the entire water column (Raateoja, 2013;

Raudsepp, 2001). With the formation of the seasonal thermocline, the CIL is formed as a separated layer between the

thermocline and the permanent halocline. Its thickness has been estimated to be 20-50 m (Liblik and Lips, 2017; Stepanova,

2017). Despite the rapid warming of the thermal mixed layer, the temperature of the CIL only slowly increases during

summer and autumn (Hinrichsen et al., 2007; Liblik and Lips, 2011). However, CIL can be traced in the water column until

the next winter (Liblik et al., 2013; Stepanova, 2017), when a new CIL is formed. The water temperature in the CIL

correlates with the severity of the previous winter (Hinrichsen et al., 2007; Liblik and Lips, 2011). After the formation of the

thermocline in spring, CIL temperature is often lower than the temperature of maximum density (Tmd), most probably due

to lateral advection of slightly higher saline, dense water. This buoyancy flux is stronger than the destabilizing effect caused

by the warming of the water, when T < Tmd (Chubarenko et al., 2017; Eilola and Stigebrandt, 1998).


According to Chubarenko and Stepanova (2018), colder and slightly saltier water, which has its origin from the upper layer
of the Bornholm Basin, advects to the east and forms the core of the CIL. Wind-driven pycnocline variations, including
coastal upwelling and downwelling events, considerably alter the depth and thickness of the CIL (Liblik and Lips, 2017). No
remarkable changes have occurred in temperature and salinity in the CIL from 1982 to 2016 (Liblik and Lips, 2019).

## 5. New knowledge of regional salinity dynamics

### 5.1 Salinity dynamics of the eastern Gotland Basin and the Gulf of Riga

The eastern Gotland Basin as part of the Baltic Proper (Fig. 1) is the most prominent region to investigate the impact of
barotropic inflows and long-term salinity changes in the Baltic Sea. The salinity dynamics of the eastern Gotland Basin is
also affecting the different sub-basins and lagoons surrounding it. The salinity dynamics there represents with sufficient
accuracy the development of salinity and stratification in the entire Baltic Proper. Changes in the mean salinity calculated
from Gotland Deep's position are only about 2% different from the calculation based on all sub-basins (Winsor et al. 2001,
Winsor et al. 2003, Elken et al. 2015).  Observed surface salinity of the eastern Gotland Basin (Fig. 2 and Fig. 4) reveals a
low-salinity period starting in the 1980s (Elken et al. 2015, Vuorinen et al., 2015, Liblik and Lips 2019) and lasting until
2002. After the MBI in 2003, the surface salinity is slightly increasing and fluctuating until 2018, but it remains relatively
low (< 6.5 g kg$^{-1}$; Fig. 4). The deepwater salinity decreased from the late 1970s until 1993 and then increased until 2018 (Fig.
2). Major saltwater inflows after 1994 can be traced by the abrupt salinity increase in the layers below the halocline. There
are also smaller barotropic inflows (Mohrholz, 2018), keeping the salinity below the halocline on a high level (Fig. 2). A
negative salinity trend of about 0.1-0.2 g kg$^{-1}$ per decade can be detected at the surface.  The surface temperature increases
by about 0.4-0.6°C per decade, whereas surface oxygen decreases by 0.1-0.2 ml l$^{-1}$ per decade (Fig. 3). Generally, the
temperature trend at the surface of the Baltic Sea follows the trend in air temperature. Increasing temperatures reduce the
solubility of oxygen, and at the same time, enhance oxygen depletion rates. Maximum negative trends in oxygen up to 1 ml
l$^{-1}$ per decade can be found in the area of the halocline (Bornholm and Gotland Basin, Fig. 3). The surface salinity trend
decreases, whereas salinity below the halocline increases (0.2-0.25 g kg$^{-1}$ per decade), leading to enhanced stratification
between the surface and deep layer. However, the frequency of barotropic and Major Baltic Inflows did not increase. The
decreasing trend in surface salinity might be due to increased runoff and net precipitation (Liblik and Lips, 2019). The
volume-averaged salinity also shows a drop until 1992, and with the MBI in 1993, a gentle increase occurred (Fig. 4 and Fig.
5).

The Gulf of Riga is a seasonally stratified, semi-enclosed basin in the eastern Baltic Sea (Fig. 1), where the water column is
fully mixed every autumn-winter. The gulf has two shallow connections with the Baltic Proper: the Irbe Strait (sill depth 25
m) and the Väinameri sea area (sill depth 5 m). The water budget in the gulf is determined by the water mass transport
through these two openings (Laanearu et al., 2000; Lilover et al., 1998; Otsmann et al., 2001) and river discharge which is



concentrated in the southern part of the gulf. Due to the shallow straits, the sub-halocline salty water does not intrude from the Baltic Proper to the gulf, and no permanent halocline exists there. Stratification in early spring is dominated by haline stratification (Stipa et al., 1999), especially close to the freshwater sources, but later in spring and summer, thermal stratification becomes more important in stabilizing the water column (Berzinsh, 1995; Liblik et al., 2017). Thus, the water

column is stratified from spring to late autumn (Berzinsh, 1995), but the mean salinity difference between the upper and deep layers is only 0.7-1.0 g kg$^{-1}$ (Raudsepp, 2001; Skudra and Lips, 2017). There is quite a high correlation between river runoff in spring and mean salinity in the upper mixed layer in August (Skudra and Lips, 2017). Bottom layer salinity in the gulf is well-correlated with the near-bottom salinity in the Irbe Strait (Skudra and Lips, 2017). Long-term changes in the average salinity are characterized by an increase from the 1960s to the late 1970s and a consecutive decrease in the 1980s-

1990s (Berzinsh, 1995). The latter trend of decreasing salinity in the gulf coincided with the corresponding changes in the Baltic Proper in the layer above the halocline during the stagnation period until the mid-1990s (Raudsepp, 2001; Fig. 2 and 4).

Wind-driven processes modify the transport of saltier water from the Irbe Strait and the advection of riverine water (Liblik et al., 2017; Lips et al., 2016b, 2016c; Soosaar et al., 2014, 2016). Most of the freshwater from the Daugava River is

transported to the north along the eastern shore during the cold season (Lips et al., 2016b). An anticyclonic gyre in the southern part of the gulf (Soosaar et al., 2014) or the entire gulf (Lips et al., 2016b) could form in spring-summer under specific wind forcing. Modeling experiments have also indicated that cyclonic eddies could develop and transport the saltier water from the Irbe Strait towards the central gulf (Lips et al., 2016c). High-resolution measurements have shown an entering of the sub-surface warmer, saltier, and oxygen-rich buoyant patches from the Irbe Strait into the gulf intermediate

layer in summer. The exact shape, fate and impact of these sub-mesoscale features are unknown, but they showed up as strong subsurface salinity maxima in the time series (Liblik et al., 2017).

### 5.1.1 Salinity dynamics of lagoons

Some of the largest European lagoons (e.g., Curonian Lagoon, Vistula Lagoon, Szczecin Lagoon) situate in the Baltic Sea (Fig. 1). As the Baltic Sea can be considered a large estuary, the Gulf of Finland, the Gulf of Bothnia and the Gulf of Riga

can be described as estuaries of medium-scale, and lagoons form the small-scale end. Common to all estuaries is the local circulation driven by the salinity difference inside and outside the estuary (Leppäranta and Myrberg, 2009). The salinity regime of lagoons is closely related to the water balance components, including river runoff, seawater inflows and intrusions, precipitation, and evaporation. All these water balance elements, as well as the air temperature, sea-surface temperature and sea level, are changing and can be expected to change in the future due to climate change in the Baltic Sea region. For

instance, the warming trend of the mean surface water temperature in the south-eastern lagoons of the Baltic Sea was 0.03 °C year$^{-1}$ in the period 1961–2008 and about 0.05 °C year$^{-1}$ after 1980 (Dailidiene et al., 2011; compare with Fig. 3).  In the Curonian Lagoon and the Vistula Lagoon, the water level rose 18 cm between 1961 and 2008, corresponding to a rate of ~4



mm year[-1] (Dailidiene et al., 2011). Furthermore, human activities such as river regulation, deepening port areas/inlets etc., can directly affect the water balance and salinity dynamics.

The Curonian Lagoon, located in the southeastern part of the Baltic Sea (Fig. 1), is the largest coastal shallow lagoon in Europe. It has a narrow connection to the Baltic Sea in the north (Klaipeda Strait with a width of 300-600 m). The lagoon receives freshwater discharge varying between 14 km$^3$ year[-1] to 33 km$^3$ year[-1] (Jakimavičius et al., 2018) with the dominant contribution from the Nemunas river. The total river runoff to the lagoon is on average about 22 km$^3$ year[-1], and it exhibits a strong seasonal pattern, peaking with snowmelt during the flood season in February-April (Jakimavičius et al., 2018).

The water in the Curonian Lagoon is theoretically exchanged in about 80 days. In the southern and central parts, which are directly influenced by river runoff, salinity is only up to 0.05 g kg[-1]. In the northern part, salinity is fluctuating between 0 and 7.5 g kg[-1]. The inflow of saline water from the Baltic Sea depends on the meteorological conditions. Furthermore, due to dredging in the transition area to the Baltic Sea, the annual mean salinity is increasing. For example, winds blowing from north and north-east may lead to an inflow of saltier waters into the lagoons. The Baltic Sea water can reach the central 305 Curonian Lagoon part even 40 km from the entrance during upwelling. Climate change projections reveal an increase in the Curonian Lagoon's salinity, linked to changes in water exchange through the Klaipėda Strait and the Nemunas runoff (Jakimavičius et al., 2018).

The Vistula Lagoon, the second largest lagoon in the Baltic Sea (Fig. 1), has an average salinity of 3.5 g kg[-1], and the salinity may vary from 0.5 g kg[-1] in the southern part up to 6.5 g kg[-1] at the Baltyisk Strait. The water balance of the Vistula Lagoon 310 was estimated by Rózyn´ski et al. (2018): yearly, 17 km$^3$ (80.2%) of water enter the lagoon through the Baltyisk Strait, riverine inflows amount to 3.62 km$^3$ (17.1%), atmospheric precipitation 0.5 km$^3$ (2.4%), evaporation 0.65 km$^3$ (3.1%), and groundwater inflows 0.07 km$^3$ (0.3%). While the Curonian Lagoon has maintained the same environmental conditions over ages, the Vistula Lagoon experienced considerable anthropogenic modification at the end of the nineteenth century, evolving from a freshwater coastal lake to an estuarine lagoon with predominant marine influence (Chubarenko et al. 2017). There are 315 plans for constructing a second inlet to the lagoon at the Polish side (Rózyn´ski et al., 2018) that could change the water balance of the lagoon in future.

The Szczecin Lagoon (Fig. 1) also belongs to one of the largest lagoons in Europe. The lagoon is shallow with an average depth of 3.8 m only. The salinity varies between 1 and 3 g kg[-1], and the lagoon is connected with the Baltic Sea via three outlets (Friedland et al., 2019). The residence time of water in the Szczecin Lagoon is about 75 days. The sea surface 320 temperature in the Curonian Lagoon is projected to increase by 2–6°C by the year 2100 (Jakimavičius et al., 2018). Water temperature and sea level rise could lead to an increase in salinity due to less restricted water exchange between the Baltic Sea and the lagoons. The average water level of the lagoons is usually higher than that of the Baltic Sea. In the future, the sea level of the Baltic Sea is projected to rise, resulting in a possible widening and deepening of connecting inlets. Thus, the water exchange between lagoons and the Baltic Proper will increase, leading to a decrease in the difference in sea-level 325 heights and an increase in salinity in the lagoons.





## 5.2 Salinity dynamics of the Gulf of Finland

The Gulf of Finland is an elongated sub-basin of the Baltic Sea located in the north-eastern extremity of the sea (Fig. 1). The length of the Gulf is about 400 km, and its width vary between 48 and 135 km (Myrberg, 1998). The mean depth of the Gulf

is 37 m, the maximum depth being 123 m (in the Baltic Sea 459 m). The drainage area of 420 990 km$^2$ is 20 % of the total drainage area of the entire Baltic Sea. The water budget in the gulf is mainly determined by unrestricted and continuous water exchange with the Northern Baltic Proper in the west and river discharge, which is mainly concentrated in the eastern part, where the largest river of the entire Baltic Sea locates; namely the river Neva. The water column can be divided into three layers – the upper mixed layer, the cold intermediate layer , and the sub-halocline near-bottom layer separated by the

seasonal thermocline and the quasi-permanent halocline, respectively (Alenius et al., 1998, 2003; Soomere et al., 2008). The seasonal thermocline vanishes on a yearly basis, during autumn-winter. Strong wind events, reversing the estuarine circulation (Elken et al., 2003), can occasionally destroy the halocline for more than a month in large areas of the gulf in winter (Liblik et al., 2013). Surface salinity increases from about 1 g kg$^{-1}$ in the easternmost part to 6 g kg$^{-1}$ in the western part. On average, surface salinity is higher near the southern coast than the northern coast (Kikas and Lips, 2016; Liblik and

Lips, 2017) due to the general cyclonic circulation scheme (Andrejev et al., 2004a, b). The westward flowing current along the northern coast is changing its location across the gulf mainly due to wind forcing (Kikas and Lips, 2016; Liblik and Lips, 2017; Lips et al., 2016a).

Wind-driven processes, such as the along-gulf advection, coupled up- and downwelling events, and vertical mixing, play an important role in the salinity dynamics. Westerly winds bring the saltier upper layer water to the gulf from the Baltic Proper

(Lilover et al., 2016; Suhhova et al., 2018), weaken the stratification and deepen the upper mixed layer (Liblik and Lips, 2017). Easterly winds have an opposite effect; these intensify the transport of fresher waters to the west (Elken et al., 2003; Liblik and Lips, 2012), shallow the mixed layer and strengthen the haline stratification (Liblik and Lips, 2017). The latter process can lead to the formation of the shallow haline stratification in winter (Liblik et al., 2013). Shallowing or deepening of the upper mixed layer due to the prevailing winds can be an important factor influencing the primary production and

species dominance during the summer cyanobacteria blooms (Kanoshina et al., 2003).

Coupled up- and downwelling events bring denser water from the cold intermediate layer to the surface layer, where it mixes with the ambient upper layer water (Myrberg and Andrejev, 2003, Lips et al., 2009). Upwelling-downwelling events in the southern/northern coast have several distinctive characteristics. Less wind forcing is needed to generate upwelling along the southern coast than the northern coast (Kikas and Lips, 2016; Liblik and Lips, 2017). Stronger lateral salinity, temperature

and density gradients occur in the upper layer in the case of upwelling along the southern coast (Kikas and Lips, 2016; Liblik and Lips, 2017). Eastward advection in the surface layer and downwelling along the southern coast generated by westerly winds can form a thick upper mixed layer (>45 m) in summer (Liblik and Lips, 2017). A positive trend in the upwelling occurrence along the northern coast was detected in 1990-2009 (Lehmann et al., 2012). However, no long-term trends were seen in the upwelling favorable winds in 1982-2013 (Liblik and Lips, 2017).





First, *in situ* measurements and modeling experiments have been conducted to characterize sub-mesoscale processes and their impact on the development of stratification and vertical mixing in the gulf (Lips et al., 2016a; Väli et al., 2013; Vankevich et al., 2016).

The gulf is impacted by estuarine circulation reversals caused by westerly wind impulses (Elken et al., 2003), which considerably weaken the halocline and lower salinity in deep layers (Elken et al., 2003; Lilover et al., 2016; Stoicescu et al.,

2019). In the case of long-lasting, strong westerly winds, circulation reversals can lead to the vanishing of the stratification in large areas of the gulf in winter (Liblik et al., 2013; Lips et al., 2017). Stratification collapse events have become more frequent since the 1990s (Elken et al., 2014). More frequent and stronger westerly winds during winters (Keevallik and Soomere, 2014) generate more reversals and likely cause salinity minimum in the annual cycle of the deep layer (Lehtoranta et al., 2017; Maljutenko and Raudsepp, 2019). The reversals, together with upward salt flux created by convective and wind

mixing, cause a maximum in the annual cycle of the upper layer salinity in the winter period. Salinity maximum/minimum usually occurs in the deep/surface layer in summer, when the seasonal thermocline restricts vertical mixing, and westerly winds are not that dominant. Another minimum in the sea surface salinity might occur due to a lack of vertical mixing in the ice-covered areas in late winter (Merkouriadi and Leppäranta, 2015).

Multi-year changes of salinity in the deep layer depend on the occurrence of MBIs (Laine et al., 2007; Liblik et al., 2018;

Liblik and Lips, 2011). If the water exchange with the North Sea was artificially limited in a numerical experiment, salinity decreased in the deep layer of the gulf (Lessin et al., 2014). After the recent MBIs (Naumann et al., 2018), salinity peaked at 10.77 g kg$^{-1}$ in the near-bottom layer of the central gulf (Liblik et al., 2018), which is the highest value since 1974 (Alenius et al., 1998). Former deep layer water from the Northern Baltic Proper was pushed to the gulf nine months after (Liblik et al., 2018) the MBI occurred in December 2014 (Mohrholz et al., 2015). The MBI water, which originates from the depths of

110-120 m in the Eastern Gotland Basin (compare with Fig. 2), arrived in the gulf 14–15 months after the inflow (Liblik et al., 2018).

Decadal trends of salinity show vertically distinct changes. Surface salinity decrease since the early 1980s has been estimated to be in the range from 0.005 g kg$^{-1}$ (Liblik and Lips, 2019) to 0.02 g kg$^{-1}$ per year (Almén et al., 2017). Long-term records close to the island of Utö revealed a sea surface salinity decrease from the early 1980s to mid-1990s (Laakso et al.,

2018; compare with Fig. 2 and Fig. 4). However, the surface salinity increased by 0.5 g kg$^{-1}$ during 1927-2012 in the north-western part of the gulf (Merkouriadi and Leppäranta, 2014). The salinity trend in the deep layer of the central gulf has been estimated to be 0.04 g kg$^{-1}$ per year from 1982-2016 (Liblik and Lips, 2019).

## 5.3 Salinity dynamics of the Gulf of Bothnia

The mean depth of the Gulf of Bothnia is 55 metres, and its surface area is 30 % of the entire Baltic Sea. The Gulf of Bothnia is mainly separated from the northern Baltic Proper by sills and archipelagos. Its hydrography is quite different from other parts of the Baltic Sea (Fig. 1). The sill between Åland Sea and Baltic Proper prevents the northward propagation of deep-



water flow. It is assumed that the water masses in the Bothnian Sea are renewed mainly by inflowing surface water from the Baltic Proper (Marmefelt and Omstedt, 1993, Meier, 2007). The net water exchange through the Archipelago Sea is
estimated to be low compared to the Åland Sea (Omstedt et al., 2004, Myrberg and Andrejev, 2006, Tuomi et al., 2018). In the Gulf of Bothnia, the salinity stratification is weak. The surface salinity in the Åland Sea is about 5.25–6.25 g kg$^{-1}$, whereas at a depth of 200 meters, salinity varies between 7 and 7.75 g kg$^{-1}$. Deep salinity in the Åland Sea and the Sea of Bothnia stems from the upper homohaline layer of the Northern Baltic Proper. Additionally, a small fraction of more saline deep-water flows in over sills. The inflow of saline water through the Åland Sea over the sills can cause a corresponding
flow of fresher water out into the Gotland Basin. This strengthens the stratification in the Sea of Bothnia (Leppäranta and Myrberg, 2009).

In the Sea of Bothnia (Fig. 1), the surface salinity varies between 4.8 and 6.0 g kg$^{-1}$ and in the lower layer at 150 m depth, the salinity is 6.4–7.2 g kg$^{-1}$. In the Bay of Bothnia (Fig. 1), the salinity is between 2 and 3.8 g kg$^{-1}$ and at 100 m depth near the bottom it varies between 4 and 4.5 g kg$^{-1}$. The Gulf of Bothnia has many rivers, and near the river mouths, the salinity is
close to zero. Even in the Sea of Bothnia, the salinity stratification is relatively weak, and overall oxygen conditions have remained fairly good, not to mention some specific coastal areas. However, oxygen conditions in the deepest layer have somewhat deteriorated over the recent two decades, although there is no real evidence that hypoxic conditions will occur in future (Raateoja et al., 2013).

## 6. Climate variability and change – impact on salinity dynamics

**6.1 Development of the mean salinity**

The long-term changes in the salinity of the Baltic Sea depend to a large extent on net precipitation, river discharge and wind forcing (Winsor et al. 2001, 2003; Meier and Kauker 2003; Gustafsson and Omstedt, 2009); higher salinity appears during dry and lower salinity during wet periods. Furthermore, on shorter timescales, the mean salinity of the Baltic Sea is governed by the water exchange between the North Sea and the Baltic Sea, which by itself is governed by the prevailing atmospheric
conditions. Generally, westerly winds force the inflow of saline water and easterly winds force outflow of brackish Baltic Sea water. Multi-decadal oscillations control the long-term variations of the surface salinity and its meridional gradient with a period of about 30 years (Radtke et al. 2020). A statistically significant positive trend of centennial changes of the north-south gradient of the surface salinity has been found (Kniebusch et al., 2019). Increased river runoff from the most northern catchment could explain this trend. Observations reveal (Fig. 5, see chapter 5.1) that after the minimum around 2003/2004,
the volume-averaged salinity increased again until 2018. Fluctuation in the accumulated anomalies of river runoff coincides with the variability in mean salinity, confirming the role of river runoff in controlling the mean salinity of the Baltic Sea (Kniebusch et al., 2019; Radtke et al. 2020).





## 6.2 Internal circulation and stratification

The long-term salinity dynamics within the Baltic Sea is controlled by the large-scale internal water cycle (e.g. Elken and Matthäus, 2008). There is a surface layer circulation and a deep-water circulation decoupled in the Baltic Proper by the permanent halocline. The wind and freshwater surplus drive the upper layer circulation. It is mainly of Ekman dynamics in combination with complex coastlines and up- and downwelling. In the lower layer, the flow, dense bottom current, is driven by internal pressure gradients steered by the complex bottom topography consisting of deep basins and channels and

restricted by sills (Leppäranta and Myrberg, 2009). The vertical branch of this circulation system, termed the Baltic Sea haline conveyer belt (Döös et al., 2004), is restricted by the strong permanent saline stratification. Convection, mechanical mixing, entrainment and vertical advection determine the vertical salt flux across the halocline. It is not only the mean salinity of the Baltic Sea which is varying over the years, there are also considerable changes in the strength of the permanent salinity stratification. Liblik and Lips (2019) found a strengthening of the permanent halocline in the deep basins

of the Baltic Sea over the period 1982-2016. They argued that the accumulated river runoff probably caused the decrease in surface salinity. However, they found no correspondence between increased runoff and decreased surface salinity in the second half of the period (compare Fig. 4 and 5). They argued that changes in the vertical salt transport might be the reason for this, which might be related to changes in meteorological forcing. However, the volume-averaged salinity of the eastern Gotland Basin is highly correlated with the accumulated river runoff (Fig. 5). After the Major Inflow in January 1993,

salinity increased in the lower layer of the eastern Gotland Basin (Fig. 2). This increasing trend of deep layer salinity (Fig. 3) due to stronger lateral salt transport from the Kattegat could not be explained by a growing number of barotropic and Major Baltic Inflows (Mohrholz, 2018). So, the reason for the change of the haline stratification in the deep basins of the Baltic Sea over the recent three decades remains unclear.

### 6.2.1 The specific role of precipitation and river runoff

River runoff (R) and net precipitation (P-E) over the sea surface are dominant drivers of the Baltic Sea salinity, explaining together with the limited water exchange with the North Sea the large gradient in sea surface salinity between about 20 g kg$^{-1}$ in Kattegat and 2 g kg$^{-1}$ in the Bothnian Bay (Leppäranta and Myrberg, 2009). Net precipitation amounts to about 10% of the total river runoff (e.g., Leppäranta and Myrberg, 2009, Meier and Döscher, 2002), even if there are some uncertainties in these estimations. For the period 1850-2008, the total river runoff from the Baltic Sea catchment area reconstructed from

observations (Hansson et al., 2011; Cyberski and Wroblewski, 2000; Mikulski, 1986; Bergström and Carlsson, 1994) and hydrological model results (Graham, 1999) show no statistically significant trend but a pronounced multi-decadal variability with a period of about 30 years (Meier et al., 2019a, Meier et al., 2019b). According to model results, these variations in runoff explained about 50% of the long-term variability of the volume-averaged salinity of the Baltic Sea (Meier and Kauker, 2003). This relationship is also confirmed for the period 1979 to 2018 presented in Fig. 5. The volume-averaged

salinity of the eastern Gotland Basin is highly negatively correlated with the accumulated anomaly of runoff to the Baltic





Sea. About 27% of the interannual salinity variation is explained by the direct dilution effect (Radtke et al. 2020). In addition to the 30 years period (Meier et al., 2018), there is a pronounced decadal variability of both mean salinity and accumulated anomaly of runoff with the minima of mean salinity directly associated with the maxima in runoff anomaly. Furthermore, minima of mean salinities occur just before major saltwater inflows (MBIs) happen. Since about the 1970s, the mean

seasonal cycle of the total river flow has changed with increasing and decreasing runoff during winter and summer (Meier and Kauker, 2003). These changes might be explained by river regulation of large rivers in the North and systematic changes in precipitation patterns due to warming in the Baltic Sea region. However, the change in seasonality does not affect the total discharge trend. As there is no statistically significant trend in saltwater inflows on the centennial time scale (Mohrholz, 2018), changes in salinity are regionally limited. Furthermore, there is neither any statistically significant long-term trend in

salinity (Fonselius and Valderrama, 2013). As a consequence of the pronounced 30-year variability in runoff and MBIs, the mean salinity shows these variations as well (Winsor et al., 2001; 2003). As part of the variability, during 1983-1993, a stagnation period without MBI and with decreasing salinity was observed (Nehring and Matthäus, 1991). Model results suggest that decreasing salinity over about 10 years appear approximately once per century on average and belongs to the natural variability of the system (Schimanke and Meier, 2016).

On longer time scales, the Baltic Sea salinity is under the influence of the AMO with a period of about 60-90 years (Börgel et al., 2018). Since about the 1980s, increased bottom and decreased surface salinities have been observed (Vuorinen et al., 2015; Liblik and Lips, 2019), and an accelerated warming might be attributed to the AMO (Kniebusch et al., 2019). Whether the recent salinity changes are connected with the AMO is still unknown.

Besides local effects on surface salinity, due to varying river runoff and net precipitation, there is an additional remote effect

due to the accumulated volume of freshwater. This water volume has to leave the Baltic Sea as a brackish surface outflow through the Danish Straits. Periods of positive anomalies of freshwater input (P-E+R) lead to increasing outflow and a shift of the wedge-shape salinity fronts in the Belt Sea and the Sound further in direction to the Kattegat, indirectly impacting the compensating inflow of higher saline water over the Drodgen and Darss Sills. During negative anomalies of freshwater input, reduced outflow occurs, and the wedge shape salinity front moves further in the direction to the Darss and Drodgen

Sills (Lehmann and Hinrichsen, 2000). As the net precipitation is in the order of 10 % of the total river runoff, wet years will lead to a decreased salt flux into the Baltic Sea, and dry years will lead to an increased salt flux.

**6.2.2 The role of sea level change due to global warming**

Especially, the northern areas experience an acceleration of global warming during recent decades. The linear trend of global temperature increase shows warming of 0.85°C over 1880 to 2012 (IPCC, 2019). The linear temperature rise for the Baltic

Sea is about 0.4°C per decade (e.g. Lehmann et al., 2011).

The absolute sea level rise of the Baltic Sea over the twentieth century is about 1.3 – 1.8 mm per year, within the range of global estimates. In more recent decades, the basin-wide range of sea level rise may be around 5 mm per year with a rather





considerable uncertainty of ± 3 mm per year, even higher than the global mean sea level (GMSL) estimate of 3.2 mm per year (Hünicke et al. 2015, Dangendorf, 2019).

Hordoir et al. (2015) investigated the influence of rising GMSL on saltwater inflows into the Baltic Sea. They performed idealized model sensitivity experiments using a regional ocean general circulation model covering the North Sea and the Baltic Sea. Hordoir et al. (2015) found a non-linear increase in saltwater inflow intensity and frequency with rising GMSL. However, their explanation of reduced mixing in the Danish Straits was shown to be wrong (Arneborg, 2016). Arneborg (2016) proposed an alternative theory instead. Due to the smaller depth, the volume flux through the Sound is more sensitive

to GMSL rise than that through the Belt Sea. Under present conditions, the amount of dense water passing the Drogden Sill in the Sound is determined by a baroclinic control in the narrow northern end of the Sound (Nielsen, 2001). With rising GMSL, this control is degraded, and relatively more saltwater is transported into the Baltic Sea compared to the expected increase when the transport change is proportional to the area of the limiting cross-section. Assuming a negligible impact of GMSL rise, the intensity and frequency of MBIs were projected to remain unchanged, with a potential tendency of a slight

increase (Schimanke et al., 2014). However, in future high-end global mean sea level projections, reinforced saltwater inflows result in higher salinity and increased vertical stratification than present conditions (Meier et al., 2017; Saraiva et al., 2019b).

## 7. The impact of salinity dynamics on the environmental conditions of the marine ecosystem

### 7.1 Oxygen conditions

Generally, the Baltic Sea's oxygen distribution results from the input through the atmosphere-ocean interface, physical transport, and the consumption of oxygen due to respiration and biogeochemical processes. The layer above the halocline is well oxygenated due to vertical convection during winter. In and below the halocline, the ventilation is happening only by horizontal advection of water originating from the Danish Straits and the Kattegat. In the deep basins below the halocline,

there is often no sufficient oxygen supply so that continuous oxygen consumption leads to hypoxic or even anoxic conditions. In the Bothnian Sea and Bothnian Bay (Fig. 1), the salinity stratification is weak, so that vertical deep convection during winter prevents these basins from oxygen deficiency so far.

Climate warming impacts not only the deep-water oxygen distribution. The solubility of oxygen at the surface depends on the water temperature. In warmer water, the solubility is reduced and, oxygen consumption is increased by enhanced

decomposition of organic matter. Thus, even after major saltwater inflows, MBIs, which can reach the deep basins of the Baltic Sea, improved oxygen conditions will faster turn back to hypoxic conditions than former times (Naumann et al. 2018). The trend in oxygen depletion is about 0.1 ml $l^{-1}$ per decade in the surface layer and up to 1 ml $l^{-1}$ per decade in the halocline (Fig. 3).





### 7.2 Environmental interaction between fish/larvae and salinity dynamics


Roughly 100 fish species occur in the Baltic Sea. Their spatial distribution is primarily governed directly by salinity and oxygen. Marine species (~ 70 species) dominate in the central Baltic Sea, while freshwater species (30-40 species) mainly occur in coastal areas and the north-eastern parts of the Baltic Sea (HELCOM, 2002). Cod, herring and sprat comprise most of the fish community in biomass and numbers. Commercially important are the marine species sprat, herring, cod, various

flatfish, salmon, and the freshwater species pike and perch before severe stock declines (Nilsson et al., 2004).

Cod eggs in the Baltic Sea have a vertical distribution concentrated in deep water and near or below the permanent halocline (Wieland and Jarre-Teichmann, 1997). Thus, cod eggs are frequently distributed in water layers with very low oxygen concentration (Nissling et al., 1994; Wieland et al., 1994). The eastern Baltic cod has a prolonged spawning period from March to September (Köster et al., 2017). The reproductive layer thickness for Eastern Baltic cod is limited to minimum

environmental threshold values (salinity = 11 g kg$^{-1}$, temperature = 1.5 °C, and oxygen 2 ml l$^{-1}$; Wieland and Jarre-Teichmann, 1997; Westin and Nissling, 1991; Nissling et al., 1994; Wieland et al., 1994). The reproductive layer thickness for western Baltic cod in February/March (the peak spawning time of this stock) is limited to minimum environmental threshold values (salinity = 16.5 g kg$^{-1}$, temperature = 2°C, and oxygen = 2 ml l$^{-1}$; Nissling and Westin, 1997) to keep western eggs in the Arkona Basin floating to allow successful development and survival. In a modeling study, Hinrichsen et

al. (2016a) showed that the spatial extent of the habitat suitable for successful fertilization and development of eggs of the eastern cod stock is primarily determined by oxygen and salinity conditions at spawning. The highest survival of Eastern Baltic cod eggs occurred in the Bornholm Basin and a pronounced temporal decrease of survival in the Gdansk Deep and the Gotland Basin. Relatively low survival in these areas was attributable to oxygen-dependent mortality. Compared to eggs spawned in these eastern spawning grounds, eggs spawned in the Arkona Basin were affected mainly by sedimentation, i.e.

the lack of sufficiently saline water at the bottom to ensure successful egg fertilization and development. However, since the mid-2000s, a substantial increase in sedimentation-related mortality has also been observed for the Gdansk Deep (Hinrichsen et al., 2016a).

For the Bornholm Basin, Hinrichsen et al. (2016b) estimated the geographic extent of the area with hydrographic conditions suitable for egg development in early (April-May), mid (June-July) and late (August-September) spawning season. Egg

survival depends on their buoyancy which is related to female age and/or size. Large eggs, for example, are spawned by large, old females and float at low water density. The seasonal timing of spawning does not matter for these large eggs, while for small eggs spawned by young females sink towards the bottom and suffer higher mortality due to

exposure to hypoxic conditions. The geographic area suitable for their survival is concurrently lower than for larger eggs, with most favorable conditions occurring late in the spawning season owing to the summer inflows.

The Arkona Basin, a relatively shallow area (max depth ~ 40 m) mainly occupied by the western Baltic cod stock, has presently also been utilized as spawning ground by the Eastern cod stock (Bleil et al., 2009; Hüssy, 2011). Generally, the





reproductive conditions appear to be more favorable for the eastern stock, with several occasions of relatively large reproductive layer thicknesses since 1999, which was extremely seldom observed for the western stock. Vertical resolution data of the reproductive layer thickness for both stocks showed improved reproductive conditions for eastern cod of about 10
m overall layer thicknesses in June to August since 2000. A certain fraction of the egg production of the Eastern cod stock can be expected to sink to the bottom and die. However, in a combined stock identification and drift model study, Hüssy et al. (2015) showed that Eastern Baltic cod progressively immigrated into the Arkona Basin during recent years. This resulted in stock mixing with the western stock, showing a marked increase in proportion between 2005 and 2008 with a fairly stable proportion of approximately 70 - 80% since then. Even though this stock mixing is purely a physical admixture without
interbreeding (Hemmer Hansen et al., 2019), the immigrating eastern cod may have contributed to the recruitment of the eastern cod stock in this management area. However, this was only possible for relatively late spawning eastern cod and in years characterized by specific conditions, i.e. after summer inflows of saline water.

## 8. Present knowledge gaps

There is still a need to understand better the role of freshwater balance on salinity distribution and its variability on seasonal to inter-annual time scales. The surface salinity of the eastern Gotland Basin varied over the recent four decades (Fig. 4). In the 1970s, it started at relatively high values of about 7.5 g kg$^{-1}$, decreased until 2002 and slightly increased until 2018. There is a pronounced interannual variability of the surface salinity, which might be related to changes in the atmospheric forcing (wind and precipitation over the sea) and/or river runoff (Radtke et al. 2020). There is still research needed to understand
better the development of salinity stratification and its role in increasing hypoxia and to evaluate the changes in atmospheric circulation and its impact on inflows and salinity distribution in the Baltic Sea. One key question is the complete mechanistic description of barotropic and major saltwater inflows, MBIs. Even if these have been studied for decades, we can question whether we really understand the process, can we predict MBIs? Extended outflow periods before the inflow reduce the mean sea level of the Baltic Sea, and in parallel lead to the formation of a haline stratification in the Danish Straits with the
highly saline water propagating to the direction of  Darss and Drodgen Sills. The frequency of low-pressure systems passing over the Baltic Sea and the strength of the wind will be enhanced for MBIs compared to LVCs. Thus, this leads to higher transport rates.

Forthcoming work needs to explore the chain of processes in detail, which additionally to large barotropic inflows leads to an influx of highly saline and oxygenated water. The freshwater input seems to play only a modulating role for the
occurrence and strength of barotropic inflows. The total frequency of inflow events will not change, but the average amount of salt which an individual event transports into the Baltic Sea (Radtke et a. 2020). Both river runoff and the strength of barotropic inflow show a variation on a 30-year timescale, and both show a stable and plausible phase relationship to be the driver of interdecadal salinity variations (Radtke et al., 2020).



Summer inflows of saline water masses can be traced in the Bornholm Basin by unusually high temperatures in the halocline
zone. Warm and salty summer inflows belong to baroclinic inflows. They might result in higher connectivity between
nursery areas of pelagic fish species west of their principal spawning grounds and spawning stocks, e.g. for Baltic cod and
flounder (Ruzzante et al., 2006; Svedäng et al., 2007).

At the time, detailed studies on summer inflows and how they might have changed over time are missing. There are also
local processes that deserve further studies. Detailed assessments of the exchange between coastal areas, including lagoons
and open sea, and between sub-basins, the cold intermediate layer and turbulent mixing are unavailable. To improve our
knowledge of these processes, we need detailed and joint modeling and observational studies.

One very topical issue, which is indirectly linked to salinity distributions, is the general circulation of the Baltic Sea. Do we
understand all branches of the Baltic Sea haline conveyer belt? There are few regular observations of it, and the modeling
exercises show significant discrepancies between their results and observations.


## 9. Key messages

The long-term salinity dynamics is controlled by river runoff, net precipitation and the governing east-west wind conditions,
i.e. the water mass exchange between the North Sea and the Baltic Sea. There is no clear long-term trend of the mean salinity
of the Baltic Sea, even if, during the last 40 years, surface salinity has decreased, and the lower layer salinity increased. This
might be connected to changes in the vertical flux of salinity, but the explanation is still unclear. Changes in runoff are
highly correlated with the development of the mean salinity of the Baltic Sea and explain about 50% of its variability (Fig.
5). A 30-year variability has been found for surface and bottom salinity, river runoff and salt transport across the Darss Sill.
Variations of salinity on shorter time scales (monthly to annual) are even more complex, especially in and below the
halocline. Furthermore, there is a direct effect on temperature and salinity distributions. Stronger saltwater inflows can
directly be traced by changes in the deep-water salinity and corresponding changes in temperature and oxygen (Fig. 2).

Over recent decades, negative salinity trends appear at the surface. At the same time, temperature increases, and oxygen
decreases. The linear trend (~0.4 °C/decade) of the sea surface temperature of the Baltic Sea is about the same as the air
temperature trend.

The decreasing trend in oxygen can partly be explained by increasing temperatures which affect oxygen solubility and
depletion rates. Maximum negative trends up to 1 ml l$^{-1}$ per decade can be found in the halocline of Bornholm and Gotland
Basin (Fig. 3).

The major saltwater inflow in December 2014 stimulated new research to revisit the barotropic water exchange between
Kattegat and the Baltic Sea.





The major saltwater inflows, MBIs, which occur in response to specific atmospheric circulation patterns, can be regarded as
a subset of barotropic inflows or Large Volume Changes.  Atmospheric forcing is more strongly associated with LVCs than MBIs, while it directly controls the sea level and indirectly the amount of salt of the inflowing water mass.

The strength of the inflows and the amount of salt transported into the Baltic Sea depend on the intensity of the wind and the haline stratification in the Danish Straits.

It has been widely speculated that MBIs play the most crucial role in the development of deepwater salinity. Still, recent studies show that the frequency of major saltwater inflows did not change.  So, the associated worsening of bottom oxygen conditions is caused by excessive nutrient loading and related oxygen consumption and maybe due to increased stratification. This strongly suggests that reducing the external nutrient load to the Baltic Sea is still highly needed to improve its ecological state.

At regional scales, in addition to the interaction with the main Baltic Sea, the salinity regime of estuaries and lagoons is closely related to the local water balance components, including river runoff, precipitation, and evaporation. So, in the changing climatic conditions, the development of the salinity regime at regional scales may have various basic-specific features that might be diverse from corresponding trends in the main Baltic Sea. This fact will raise a high demand to carry out basin-specific studies to understand the changes in the local salinity regime.

Finally, we can summarize our present knowledge of salinity dynamics as follows. There is now a better overall view of the salinity of the Baltic Sea than before: not only the fragments in various scales of time and space are known. The measurements have given an improved view also in the regional scale, in gulfs and lagoons, not only in the central part of the sea, or some volume-averaged case. Future projections are existing, but they still have lot of uncertainties. The temporal variability can now been divided between the decadal and smaller scales.


**Author contribution**

Andreas Lehmann wrote the main parts of the manuscript and prepared the manuscript with contributions from all co-authors. Kai Myrberg contributed to the main parts of the manuscript. Piia Post also contributed to the overall manuscript and wrote the main parts of chapter 3. Irina Chubarenko contributed to chapter 4. Inga Dailidiene contributed to chapter 5.
Hans-Harald Hinrichsen contributed to chapter 7. Karin Hüssi also contributed to chapter 7. Taavi Liblik commented on the entire draft manuscript and contributed to the main parts of chapter 4 and 5. Urmas Lips also contributed to the main parts of chapter 4 and 5. Markus Meier commented on the entire manuscript and contributed to chapter 6, and Tatiana Bukanova contributed to chapter 4.



**Code availability**

Not applicable.

**Data availability**

Not applicable.

**Competing interests**

The authors declare that they have no conflict of interest.

Acknowledgements

This work has been initiated and accompanied by Baltic Earth. We are incredibly grateful for the support of the Baltic Earth secretariat.

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



**Figures**


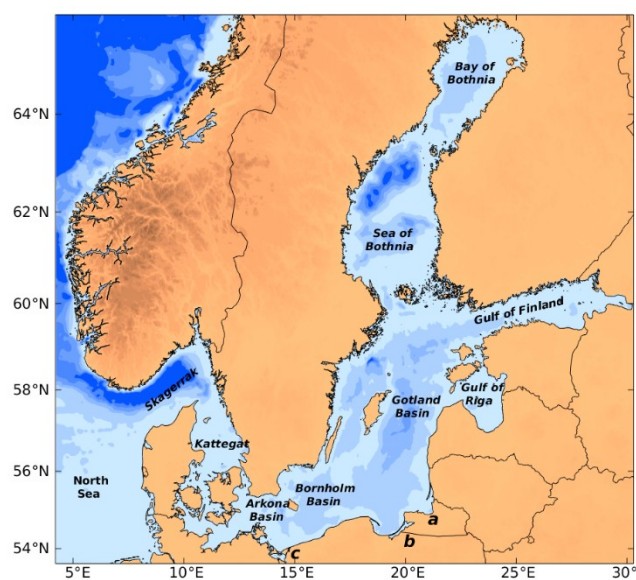



Figure 1: Map of the Baltic Sea and its sub-basins. The Baltic Proper comprises the sub-basins: Arkona, Bornholm and Gotland Basin. The Gulf of Bothnia comprises the Sea of Bothnia and the Bay of Bothnia. (a), (b) and (c) denote the Curonian, Vistula and Szczecin Lagoons.









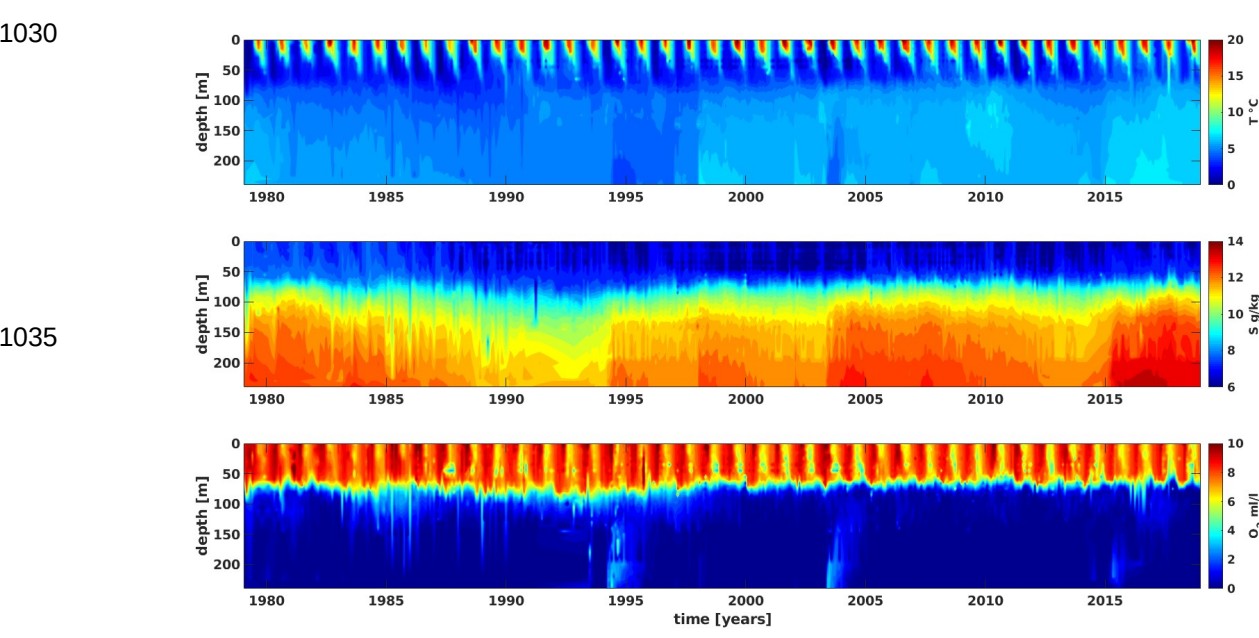


Figure 2. Time series of temperature (top), salinity (middle) and oxygen (bottom) of ICES profiles of Subdivision (SD) 28 (eastern Gotland Basin) for the period 1979-2018.








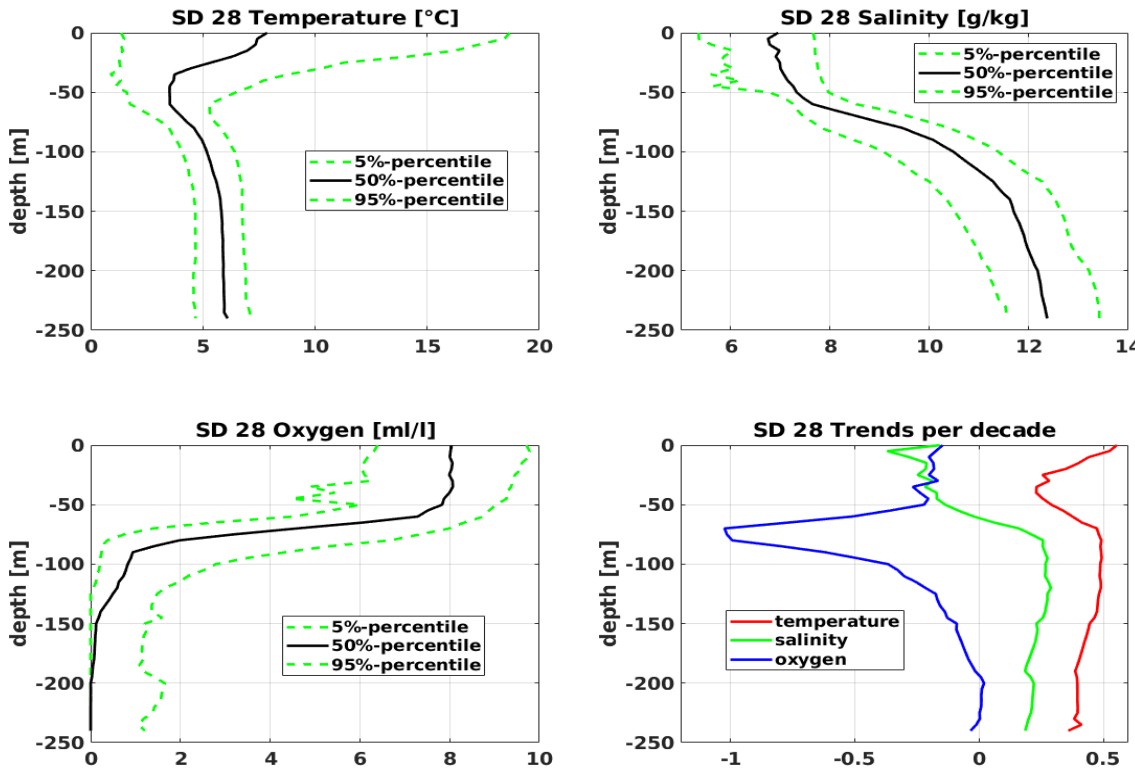

**Figure. 3. Percentiles (5, 50 and 95%) of temperature, salinity and oxygen profiles for SD 28 (eastern Gotland Basin) for the period 1979-2018. Trends per decade of temperature, salinity and oxygen based on SD 28 temperature, salinity and oxygen profiles for the period 1979-2018 (right lower panel).**







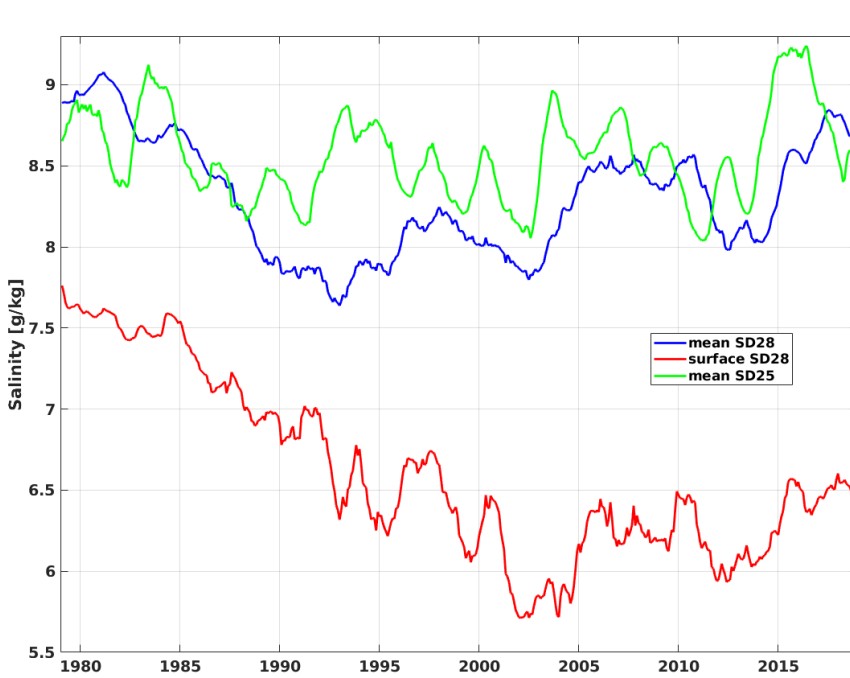

**Figure 4. Mean salinity of SD 28 (eastern Gotland Basin) (blue) and surface salinity of SD 28 (red). For comparison mean salinity of SD 25 (Bornholm Basin) (green) for the period 1979-2018. All series 12 months running mean.**



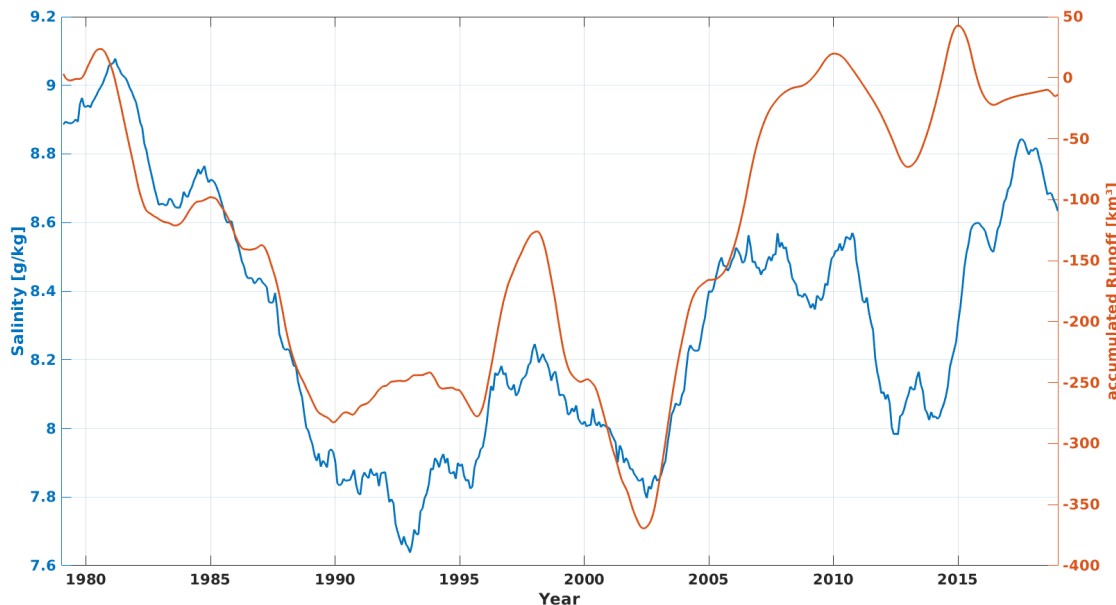

1110  **Figure. 5. Volume averaged salinity of SD28 for the period 1979 to 2018 and accumulated anomalies of runoff to the Baltic Sea (inverted). The correlation coefficient is 0.75. All series 12 months running mean.**

1115