# Peer review of "Salinity dynamics of the Baltic Sea"

_Earth System Dynamics, 2021_

## Referee Comment (RC1)

Submitted to Earth System Dynamics esd-2021-15

**General comments**

The MS presents an interesting review of recent advances in Baltic Sea physical oceanography, from the viewpoint how different physical processes affect the variability of salinity. The review goes beyond the traditional focus on Major Baltic Inflows and their reflections in different deep basins, up to the Gotland Deep. The MS considers also salinity variations and key processes behind them in the more river-influenced sub-basins (Gulf of Finland, Gulf of Riga, Gulf of Bothnia, lagoons), variations in mean salinity, and specific features like effect of salinity-dependent temperature of minimum density. Regarding external influences, recent findings of atmospheric forcing and of terrestrial freshwater flux are reviewed. Response of salinity depends on the circulation and stratification that are analyzed as well. Possible impact of global sea level rise on Baltic salinity is discussed. Finally, Baltic-specific effects of varying salinity on marine ecosystem (lateral advection of deep oxygen-rich water, salinity-dependent cod spawning volume and other fisheries issues) are considered.

The MS refers to the earlier studies within two phases of the Assessment of Climate Change for the Baltic Sea Basin (BACC I and BACC II) and some other review publications. As a good start for the updated review, earlier findings concerning salinity dynamics have been summarized in a well-readable compact form. The MS contributes to the Baltic Earth program activity of Assessment Reports (BEAR) summarizing the studies conducted around the Baltic Earth strategic research challenge under the same name as the title of MS.

The MS is comprehensive and well written. It could be endorsed for publication with some technical corrections only. Still I give below some specific comments for consideration by the authors, suggesting how the MS might be improved.

**Specific comments on sub-chapters**

Well-written sub-chapter 3. Atmospheric forcing driving the salinity dynamics of the Baltic Sea contains an ambiguity regarding LVCs and MBIs. They are treated here in combination (LVCs/MBIs), earlier they have been mentioned separately. Their detailed explanation comes later in sub-chapter 4.1, but until that, readers (especially students) are left without any short guidance what they have in common and what are the differences. Even a forward citation to 4.1 could help.

If said the above, I found that 4.1. Large Volume Changes and Major Baltic Inflows lacks condensed explanations for the recently introduced terms like revised MBI and LVC in reference to traditional MBI. I think the wider marine research community could be interested to read such information in the review paper.

Lines 199-201 contain the statement "There are also smaller inflows of barotropic origin. These occur during all seasons having a low variability between the years. Such inflows bring about 30 % of the entire salt transport to the Baltic Sea". It could be interesting to know on what depth levels the waters from smaller inflows occur and how much the earlier findings agree with the more recent results. For example, Meier and Kauker (2003) and Meier (2005) estimated that east from the Gotland transport of more saline waters incoming from the Bornholm Basin is concentrated on about 100 m depth, just below the halocline but clearly above the bottom layers.

In 4.2. The cold intermediate layer there is a statement "colder and slightly saltier water, which has its origin from the upper layer of the Bornholm Basin, advects to the east and forms the core of the CIL"(L229-230). To my knowledge, Bornholm basin as the origin of CIL (without specifying the region) is an interesting hypothesis and should be presented in a milder form, like "It has been proposed...". For example, CIL is formed also in the Gulf of Finland, about 1000 km away from the Bornholm Basin. It is not easy to understand, how this water is directly advected from the remote area.

6.1. Development of the mean salinity. The approach of volume-average salinity is an appropriate indicator for ongoing and projected changes. It might be reminded for the wider marine research audience, how much of the volume and area fraction cover the deep quasi-stagnant regions with depth more than 150 m, which are quite often put on the focus in the point measurements.

6.2.1. The specific role of precipitation and river runoff. In the nice overview, surface salinity response to seasonal maximum of river discharge during spring could be outlined as well. For example, lines 370-372 say about the Gulf of Finland "Salinity maximum/minimum usually occurs in the deep/surface layer in summer, when the seasonal thermocline restricts vertical mixing, and westerly winds are not that dominant." Seasonal course of freshwater supply is ignored in this discussion.

Climate change causes among other effects also change in timing of floods and maximum river flows (Blöschl et al., 2017). It could be interesting to find a discussion how it might affect the salinity dynamics and related ecosystem response of the Baltic Sea. Earlier in the MS there are already references on stratifying effects of the "juvenile freshwater" in the Baltic Proper by Eilola and Stigebrandt (1998) and in the Gulf of Riga by Stipa et al. (1999), Liblik et al. (2017) and Skudra and Lips, (2017). Seasonal freshwater fluxes in the Baltic have been evaluated by Hordoir and Meier (2010).

7. The impact of salinity dynamics on the environmental conditions of the marine ecosystem. Presently this sub-chapter considers mainly the issues of ecological dynamics of deep-water, like deep oxygen deficiency and cod spawning volume. "Horohalinicum" critical salinity range of 5-7 g kg$^{-1}$ that separates the freshwater and marine species population areas (Vuorinen, 2015), could be iterated as well in this sub-chapter.

**Technical remarks**

Line 32: Abbreviation BACC is introduced without explanation

Line 36: Abbreviation MBI is introduced without explanation

L50: The sentence "Despite this long research history, there are still gaps in our knowledge of salinity changes, both in space and time" could be reformulated. There will always be knowledge gaps; when some gaps get reasonably clear then other gaps will emerge.

L87: Instead of "more extensive" perhaps "larger".

L89: "lack of inflows" needs refinement, perhaps "lack of inflows reaching the deepest layers".

L91-94: "However, despite the abrupt changes in salinity, there was no clear trend for the vertical mean salinity" could need refinement as well. Is it "abrupt changes in deep salinity". What is the "vertical mean salinity"? If it is total salt amount in a basin divided by the volume of the basin, then it could be named "mean salinity over basin".

L98: "there should be firstly winds from the east, and after such a period," it could be nice to read about duration and speed of easterly winds, required for MBI.

L105: "Summer inflows inject higher saline water with higher temperatures and low oxygen content into the halocline" needs refinement. There are occasionally larger baroclinic inflows that reach the bottom layer of the Gotland Deep, bringing warm and low-oxygen water. See https://helcom.fi/wp-content/uploads/2020/07/BSEFS-Water-exchange-between-the-Baltic-Sea-and-the-North-Sea-and-conditions-in-the-deep-basins-2017.pdf

L171: "A critical study objective is the transition area", I think an area is object not objective.

L204-206: "Holtermann et al. (2017) investigated the dynamics of the deep waters and vertical mixing in the central part of the Baltic Sea while Major Baltic Inflows took place; thus, providing new information on dense bottom gravity currents on their way to the deep central Baltic Sea and associated turbulent mixing.": it could be interesting to read (at least with a few words) what is the essence of the obtained new information.

L207: "Liblik et al. (2018) studied the impact of MBIs downstream from the eastern Gotland Basin to the Gulf of Finland.": also, what are the main results.

L360: Start of the sentence is not clear.

L391: "The Gulf of Bothnia is mainly separated from the northern Baltic Proper by sills and archipelagos", why it is separated only mainly, what are then non-main separations? Perhaps the Gulf of Bothnia is connected to the northern Baltic Proper by …

L520: "Environmental interaction between fish/larvae and salinity dynamics", interaction is not correct.

Fig. 2. Subdivision 28 (SD 28) must be defined with bounds here and on later figures.

Fig. 4. SD 25 must be defined with bounds.

**References**

Blöschl, G., Hall, J., Parajka, J., Perdigão, R.A., Merz, B., Arheimer, B., Aronica, G.T., Bilibashi, A., Bonacci, O., Borga, M. and Čanjevac, I., 2017. Changing climate shifts timing of European floods. *Science*, *357*(6351), pp.588-590.

Hordoir, R. and Meier, H.M., 2010. Freshwater fluxes in the Baltic Sea: A model study. *Journal of Geophysical Research: Oceans*, *115*(C8).

Meier, H.M., 2005. Modeling the age of Baltic Seawater masses: quantification and steady state sensitivity experiments. *Journal of Geophysical Research: Oceans*, *110*(C2).

July 18, 2021

---

## Author Response (AR1)

**RC1**

**Review of MS „Salinity dynamics of the Baltic Sea" by Andreas Lehmann et al.**

**Submitted to Earth System Dynamics esd-2021-15**

**General comments**

The MS presents an interesting review of recent advances in Baltic Sea physical oceanography, from the viewpoint how different physical processes affect the variability of salinity.

The review goes beyond the traditional focus on Major Baltic Inflows and their reflections in different deep basins, up to the Gotland Deep. The MS considers also salinity variations and key processes behind them in the more river-influenced sub-basins (Gulf of Finland, Gulf of Riga, Gulf of Bothnia, lagoons), variations in mean salinity, and specific features like effect of salinity-dependent temperature of minimum density. Regarding external influences, recent findings of atmospheric forcing and of terrestrial freshwater flux are reviewed. Response of salinity depends on the circulation and stratification that are analyzed as well. Possible impact of global sea level rise on Baltic salinity is discussed. Finally, Baltic-specific effects of varying salinity on marine ecosystem (lateral advection of deep oxygen-rich water, salinity-dependent cod spawning volume and other fisheries issues) are considered.

The MS refers to the earlier studies within two phases of the Assessment of Climate Change for the Baltic Sea Basin (BACC I and BACC II) and some other review publications. As a good start for the updated review, earlier findings concerning salinity dynamics have been summarized in a well-readable compact form. The MS contributes to the Baltic Earth program activity of Assessment Reports (BEAR) summarizing the studies conducted around the Baltic Earth strategic research challenge under the same name as the title of MS.

The MS is comprehensive and well written. It could be endorsed for publication with some technical corrections only. Still I give below some specific comments for consideration by the authors, suggesting how the MS might be improved.

**Specific comments on sub-chapters**

Well-written sub-chapter 3. Atmospheric forcing driving the salinity dynamics of the Baltic Sea contains an ambiguity regarding LVCs and MBIs. They are treated here in combination (LVCs/MBIs), earlier they have been mentioned separately. Their detailed explanation comes later in sub-chapter 4.1, but until that, readers (especially students) are left without any short guidance what they have in common and what are the differences. Even a forward citation to 4.1 could help. If said the above, I found that 4.1. Large Volume Changes and Major Baltic Inflows lacks condensed explanations for the recently introduced terms like revised MBI and LVC in reference to traditional MBI. I think the wider marine research community could be interested to read such information in the review paper.

**Lines 199-201** contain the statement "There are also smaller inflows of barotropic origin.These occur during all seasons having a low variability between the years. Such inflows bring about 30 % of the entire salt transport to the Baltic Sea". It could be interesting to know on what depth levels the waters from smaller inflows occur and how much the earlier findings agree with the more recent results. For example, Meier and Kauker (2003) and Meier (2005) estimated that east from the

Gotland transport of more saline waters incoming from the Bornholm Basin is concentrated on about 100 m depth, just below the halocline but clearly above the bottom layers.

**In 4.2. The cold intermediate layer** there is a statement "colder and slightly saltier water, which has its origin from the upper layer of the Bornholm Basin, advects to the east and forms the core of the CIL"(L229-230).

To my knowledge, Bornholm basin as the origin of CIL (without specifying the region) is an interesting hypothesis and should be presented in a milder form, like "It has been proposed...". For example, CIL is formed also in the Gulf of Finland, about 1000 km away from the Bornholm Basin. It is not easy to understand, how this water is directly advected from the remote area.

**6.1. Development of the mean salinity**. The approach of volume-average salinity is an appropriate indicator for ongoing and projected changes. It might be reminded for the wider marine research audience, how much of the volume and area fraction cover the deep quasi-stagnant regions with depth more than 150 m, which are quite often put on the focus in the point measurements.

**6.2.1. The specific role of precipitation and river runoff.** In the nice overview, surface salinity response to seasonal maximum of river discharge during spring could be outlined as well. For example, lines 370-372 say about the Gulf of Finland "Salinity maximum/minimum usually occurs in the deep/surface layer in summer, when the seasonal thermocline restricts vertical mixing, and westerly winds are not that dominant." Seasonal course of freshwater supply is ignored in this discussion. Climate change causes among other effects also change in timing of floods and maximum river flows (Blöschl et al., 2017). It could be interesting to find a discussion how it might affect the salinity dynamics and related ecosystem response of the Baltic Sea. Earlier in the MS there are already references on stratifying effects of the "juvenile freshwater" in the Baltic Proper by Eilola and Stigebrandt (1998) and in the Gulf of Riga by Stipa et al. (1999), Liblik et al. (2017) and Skudra and Lips, (2017). Seasonal freshwater fluxes in the Baltic have been evaluated by Hordoir and Meier (2010).

**7. The impact of salinity dynamics on the environmental conditions of the marine ecosystem.** Presently this sub-chapter considers mainly the issues of ecological dynamics of deep-water, like deep oxygen deficiency and cod spawning volume. "Horohalinicum" critical salinity range of 5-7 g kg-1 that separates the freshwater and marine species population areas (Vuorinen et al., 2015), could be iterated as well in this sub-chapter.

**Technical remarks**

Line 32: Abbreviation BACC is introduced without explanation

Line 36: Abbreviation MBI is introduced without explanation

L50: The sentence "Despite this long research history, there are still gaps in our knowledge of salinity changes, both in space and time" could be reformulated. There will always be knowledge gaps; when some gaps get reasonably clear then other gaps will emerge.

L87: Instead of "more extensive" perhaps "larger".

L89: "lack of inflows" needs refinement, perhaps "lack of inflows reaching the deepest layers".

L91-94: "However, despite the abrupt changes in salinity, there was no clear trend for the vertical mean salinity" could need refinement as well. Is it "abrupt changes in deep salinity". What is the

"vertical mean salinity"? If it is total salt amount in a basin divided by the volume of the basin, then it

could be named "mean salinity over basin".

L98: "there should be firstly winds from the east, and after such a period," it could be nice to read

about duration and speed of easterly winds, required for MBI.

L105: "Summer inflows inject higher saline water with higher temperatures and low oxygen content

into the halocline" needs refinement. There are occasionally larger baroclinic inflows that reach the

bottom layer of the Gotland Deep, bringing warm and low-oxygen water. See https://helcom.fi/wpcontent/uploads/2020/07/BSEFS-Water-exchange-between-the-Baltic-Sea-and-the-North-Sea-andconditions-in-the-deep-basins-2017.pdf

L171: "A critical study objective is the transition area", I think an area is object not objective.

L204-206: "Holtermann et al. (2017) investigated the dynamics of the deep waters and vertical mixing in the central part of the Baltic Sea while Major Baltic Inflows took place; thus, providing new information on dense bottom gravity currents on their way to the deep central Baltic Sea and associated turbulent mixing.": it could be interesting to read (at least with a few words) what is the essence of the obtained new information.

L207: "Liblik et al. (2018) studied the impact of MBIs downstream from the eastern Gotland Basin to the Gulf of Finland.": also, what are the main results.

 L360: Start of the sentence is not clear.

L391: "The Gulf of Bothnia is mainly separated from the northern Baltic Proper by sills and archipelagos", why it is separated only mainly, what are then non-main separations? Perhaps the Gulf of Bothnia is connected to the northern Baltic Proper by …

L520: "Environmental interaction between fish/larvae and salinity dynamics", interaction is not

correct.

Fig. 2. Subdivision 28 (SD 28) must be defined with bounds here and on later figures.

Fig. 4. SD 25 must be defined with bounds.

**References**

Blöschl, G., Hall, J., Parajka, J., Perdigão, R.A., Merz, B., Arheimer, B., Aronica, G.T., Bilibashi, A., Bonacci, O., Borga, M. and Čanjevac, I., 2017. Changing climate shifts timing of European floods. Science, 357(6351), pp.588-590.

Hordoir, R. and Meier, H.M., 2010. Freshwater fluxes in the Baltic Sea: A model study. Journal of Geophysical Research: Oceans, 115(C8).

Meier, H.M., 2005. Modeling the age of Baltic Seawater masses: quantification and steady state sensitivity experiments. Journal of Geophysical Research: Oceans, 110(C2).

July 18, 2021

**Reply to 'General comments'**

Thank you for your the comments.

**Reply to 'Specific comments on sub-chapters'**

**LVCs/MBIs**
A very good comment. We will give a detailed explanation of the difference between LVCs and MBIs as earliest as possible in the manuscript.

**Smaller barotropic inflows**
The depth level on which smaller barotropic inflows occur depends on the salinity and volume of the inflowing water mass, and on the salinity distribution in and below the halocline.  On its way through Arkona, Bornholm and Gotland Basin the water is diluted and mixed by ambient water. If the Bornholm Basin is already filled up by higher saline water due to earlier inflows, the water slip up and  pass the Bornholm basin quickly,  and enter the Gotland Basin stratified in the halocline. So the water can interleave in or below the halocline in dependency on its initial salinity and mixing conditions. This process is described with respect to oxygenation of the central Baltic Sea deep Basins for example in Holtermann et al. (2017) and  Neumann et al. (2017). We will take that up in the revised manuscript.

**The cold intermediate layer:** Sorry, this formulation is unclear.  It is the water from the southwest (surface of the Bornholm Basin) which  is advected to intermediate layers in the south-east (Baltic Proper). We will clarify this in the revised manuscript.

**Development of the mean salinity**. We will give an estimate of the volume related to the depth range from 150 m  to the bottom, and the corresponding area fraction in the revised manuscript.

**The specific role of precipitation and river runoff:** We will discuss the annual cycle of freshwater and its effect on surface salinity. Thank you for the references. We will include them in the revised version.

**The impact of salinity dynamics on the environmental conditions of the marine ecosystem**

Thank you, very good comment. We will introduce a short discussion based on Vuorinen et  al. (2015) on ecological consequences to changing salinity conditions in the Baltic Sea.

**Reply to 'Technical remarks'**

Thank you for the valuable remarks. We will consider them all in the revised version.

**RC2**

The paper is a fine overview of the state of knowledge and the lack thereof concerning salinity in the Baltic Sea.

1. Science: claims of periodicity appear not warranted – instead the term 30-year variability as used in line 603 is adequate. Also, the reference to "statistically significant" trends seems a little liberal – which time windows are tested, which autocorrelation has been assumed? What does "significance" of a trend really mean – that it can be discriminated from other trends related to (which?) low-frequency variability?
2. The text does not discuss the role of small-scale variability, such as eddies and frontal dynamics – given the attention introduced by the Nobel prize, one wonders if these small scale "noise" features may give rise to low-frequency variations aka "stochastic climate model"? If nobody has thought about this issue, it should be mentioned under "gaps of knowledge".
3. Style: Very many details are presented – why not opening each section with a short (5 lines?) introductory summary of what the reader will find in the section?
4. In Section 6, a reference to the sea level-text of the BEAR family may be useful.

Thank you very much for your valuable comments

Reply to 1. Science: In this section salinity variability of different time scales is somehow mixed which might cause some confusion. We rearrange this chapter starting from long-term variability over decades (Kniebusch et al. 2019; Radtke et al. 2020) to annual and monthly changes. A linear trend could not be found over the 35-year period 1982 to 2016 (Liblik and Lips (2019), see also (Windsor, 2001 and Fonselius and Valderrama 2003).

Reply to 2. A good point, the dynamics of small scale variability eddies and frontal regions are highly important for the salinity dynamics in total. We are not aware of recent studies since BACC II focusing on small-scale variability affecting low-frequency variations in the Baltic Sea. So, we will take this up in the chapter as knowledge gap.

Reply 3. We take this point under consideration and will check whether a short introductory summary at the beginning of each chapter will improve reading the manuscript without introducing too much repetitions. In the introduction, we give a short overview of the content of each chapter. Where appropriate we added a sentence.

4. Good comment, we will definitely refer to Weise et al. (2021).

References

Weisse, R., Dailidiene, I., Hünicke, B., Kahma, K., Madsen, K., Omstedt, A., Parnell, K., Schöne, T., Soomere T., Zhang, W., Zorita, E.: Sea level dynamics and coastal erosion in the Baltic Sea region. Earth Syst. Dynam., 12, 871-898, 2021. HTTPS://DOI.ORG/10.5194/ESD-12-871-2021